# An Improved Blind Zone Channelization Structure and Rapid Implementation Method

**DOI:** 10.3390/mi14051091

**Published:** 2023-05-22

**Authors:** Ziliang Jia, Hongxia Liu

**Affiliations:** Key Laboratory for Wide Band Gap Semiconductor Materials and Devices of Education Ministry, School of Microelectronics, Xidian University, Xi’an 710071, China; 20111223127@stu.xidian.edu.cn

**Keywords:** digital channelization, polyphase filtering, joint decision, high level synthesis

## Abstract

The paper proposes an enhanced design for broadband digital receivers that aims to improve signal capture probability, real-time performance, and the hardware development cycle. To overcome the issue of false signals in the blind zone channelization structure, this paper introduces an improved joint-decision channelization structure that reduces channel ambiguity during signal reception. Xilinx’s high-level synthesis (HLS) tools are used for accelerated algorithm implementation, and techniques such as pipelining and loop parallelization are employed to reduce system latency. The entire system is implemented on FPGA. The simulation results demonstrate that the proposed solution effectively eliminates channel ambiguity, improves algorithm implementation speed, and meets the design requirements.

## 1. Introduction

Since the beginning of the 21st century, with the rapid development of modern electronic technology, electronic information warfare has become a major form of warfare between nations. In electronic warfare equipment, more and more advanced technologies are being applied in the fields of jamming, anti-jamming, reconnaissance, and counter-reconnaissance, making the electromagnetic environment increasingly complex [1,2]. At the same time, the radio signals generated by these devices also vary in signal energy and carrier frequency size. To adapt to such a complex environment, electronic warfare receivers need to have a wide monitoring frequency band range, real-time signal processing, a large dynamic range, and the ability to process time–domain overlapping signals [3,4,5].

Traditional electronic warfare receivers mostly use analog technology design, including crystal video receivers, compression receivers, superheterodyne receivers, Bragg receivers, instantaneous frequency measurement (IFM) receivers [6], and channelized receivers. Crystal video receivers have the advantage of a simple structure but can only be used for signal detection and cannot measure signal frequency. Compression receivers use the Fourier transform to compress input signals of different frequencies into time–domain pulse signals, which has good data processing performance, but signal compression may lead to system detection errors. Superheterodyne receivers typically use multi-stage mixing to convert the input RF (radio frequency) signal to intermediate frequency (IF) for processing at a lower frequency, with a good reception dynamic range, but requiring the introduction of filters to eliminate local oscillator leakage, increasing the receiver cost and complexity. Bragg receivers convert input signals into optical signals through an optical Bragg cell [7], and use a small amount of hardware resources to realize a large number of channels, but have a small reception dynamic range and low sensitivity. IFM receivers mainly use the non-linear characteristics of crystal detectors to measure the frequency of input signals, which can cover a larger monitoring frequency range, with high frequency measurement accuracy, but poor frequency measurement accuracy for time–domain overlapping signals, making it difficult to meet the actual electronic warfare requirements. Channelized receivers divide the working frequency band into multiple channels and, after channelization processing, signals of different frequencies appear in different channels with a large monitoring bandwidth and reception sensitivity, capable of processing time–domain overlapping signals; these receivers can meet the actual electronic warfare requirements and thus have important research value.

Due to the limitations of analog devices, traditional receivers have a good performance but are heavy and bulky, with poor flexibility [8]. With the rapid development of electronic information technology, more and more digital signal processing methods have been applied in the field of digital receivers. At the same time, with the continuous breakthrough of high-speed ADC (analog-to-digital converter) sampling technology, its sampling rate has been increased from MHz to GHz, enabling RF signals to be directly sampled, which also puts higher demands on backend data processing [9,10].

The ability to achieve full probability reception of signals across the entire frequency band is the most important performance metric for wideband digital receivers. A simple and effective method is to construct an overlapped digital channelization model, where the channel parts overlap and the processed signals are seamlessly joined in the frequency domain [11,12]. Although this method can achieve blind-spot-free signal reception, there is some wasting of the frequency spectrum due to the overlapping of the passbands. Another method is to use interpolated half-band filters to decompose the input signal into two complementary and non-overlapping channel signals in the frequency spectrum, and then filter each channel signal with an analysis filter bank to achieve channelization [13]. Although this method can correctly receive wideband channelized signals, its complex structure increases the difficulty of hardware implementation. Based on the limitations of the above methods, this article uses a 50% overlapping filter bank partitioning method [14] to construct filters and proposes an improved decision algorithm to eliminate the introduced ambiguity between channels. This method can avoid wasting spectral resources and is more suitable for hardware implementation. After introducing the decision algorithm proposed in this paper, it can effectively improve the signal acquisition capability of the receiver and improve the resolution of the output signal.

The limited application of broadband digital receivers is partially due to the mismatch between the speed of hardware development and theoretical development. With the rapid development of integrated circuit technology, digital receivers based on field-programmable gate arrays (FPGAs) have a good performance and can meet practical needs [15]. FPGAs have a relatively smaller size and weight compared with clusters of computers, multi-core processors, and many-core processors, etc. [16].

In the hardware implementation of the digital channelization receiver, if the traditional FPGA development process is used, the debugging and development cycle of the entire system will also increase when the data volume is large. Refs. [17,18] have both used the traditional FPGA development process, and when the number of channels is large, the delay of each part will also increase, leading to unreasonable timing in the system. In this article, a new development tool, the HLS development platform, is used in combination. To address problems in traditional FPGA development processes, such as long development cycles and tedious debugging processes, this tool uses the C/C++ high-level programming language to design and implement algorithms, and synthesizes the designed program into Verilog language to complete the transformation to the RTL level [19]. In the synthesis process, optimization instructions such as pipeline and loop parallelism can be added to reduce system delay. By combining the HLS platform with the traditional development process, the hardware implementation efficiency and design flexibility of the entire receiver system can be greatly improved.

This paper presents an improved method to address the issues of false signals in digital receivers. The proposed method transforms the process of eliminating false signals into extracting amplitude-frequency information from the output signals. This not only improves the discrimination of useful signals, but also simplifies the implementation process of FPGAs. Additionally, the paper employs a hybrid design approach by combining the HLS design approach [19] with traditional methods to accelerate the implementation process of front-end algorithms, resulting in increased hardware implementation efficiency and design flexibility for the entire receiver system.

## 2. Polyphase Filtering Digital Channelization Model

### 2.1. The Original Structure

For the digital channelization filter bank, the input signal s(n) is down-converted and then evenly divided into *K* subbands by *K* filters. After *D*-fold decimation, the output results are obtained, and this process satisfies:(1)K=F⋅D

When *F* = 1, it becomes a critical decimation process, and the original structure of the channelized receiver is shown in Figure 1.

The input signal is first modulated and then shifted to baseband. Afterward, it undergoes low-pass filtering and decimation before being fed into subsequent modules for further processing. However, at this stage, the filtering operation occurs before decimation, leading to redundant computations during signal processing. Moreover, convolution operations must be completed within a single sampling period. To address these issues, an alternative approach is proposed, which involves swapping the positions of filtering and decimation operations. This way, the computational requirements of various modules in the system can be met more efficiently.

### 2.2. Subchannelization

Multi-rate signal processing with the polyphase filtering technique plays a crucial role in reducing the complexity of signal rate conversion and lowering system design complexity. The key process in this technique is subchannelization, where the target monitoring frequency band is divided according to certain rules, and different partitioning methods correspond to different system complexities, computational loads, and resource consumption. In this design, the filter group adopts a 50% overlapping partitioning method [20], as shown in Figure 2, which concatenates the passbands of adjacent subchannels, eliminating any blind spots in the entire monitoring bandwidth and achieving the full probability of interception.

### 2.3. Improved Non-Blind Channelization Structure

The DFT-based polyphase channelization algorithm is developed based on the channelization structure of the low-pass filter bank. Given the input signal *s*(*n*), the output $f_{k}\left(m\right)$ of the kth subchannel is defined as:(2)$$\begin{array}{ll} y_{k}(m) & ={\left\{[s(n)e^{j\omega_{k}n}]*h(n)\}\right|}_{n=mD}\\ & ={\left\{\sum_{i=-\infty }^{+\infty }s(n-i)e^{j\omega_{k}(n-i)}\cdot h(i)\right\}|}_{n=mD}\\ & =\sum_{i=-\infty }^{+\infty }s(mD-i)e^{j\omega_{k}(mD-i)}\cdot h(i)\\ & =\sum_{p=0}^{D-1}\sum_{i=-\infty }^{+\infty }s(mD-iD-p)e^{j\omega_{k}(mD-iD-p)}\cdot h(iD+p) \end{array}$$

Now, let $s_{p}\left(m\right)=s\left(mD-p\right)$ and $h_{p}\left(m\right)=h\left(mD+p\right)$. Then, we can further simplify Equation (2) as follows:(3)$$y_{k}(m)=\sum_{p=0}^{D-1}\left[\sum_{i=-\infty }^{+\infty }s_{p}(m-i)e^{j\omega_{k}(m-i)D}\cdot h_{p}(i)\right]e^{-j\omega_{k}p}$$

Convert part of the relationship in Equation (3) to:(4)$$x_{p}(m)=\sum_{i=-\infty }^{+\infty }s_{p}(m-i)e^{j\omega_{k}(m-i)D}\cdot h_{p}(i)=\left[s_{p}(m)e^{j\omega_{k}mD}\right]*h_{p}(m)$$

The center frequency of each subchannel in the filter bank under the even-indexed arrangement is:(5)$$\omega_{k}=\frac{2\pi k}{D}$$

Substituting Equation (5) into Equation (4), we obtain:(6)$$x_{p}(m)=\left[s_{p}(m)e^{jm\frac{2\pi k}{D}\cdot D}\right]*h_{p}(m)=s_{p}(m)*h_{p}(m)$$

Substituting Equation (6) into Equation (3), we obtain:(7)$$y_{k}(m)=\sum_{p=0}^{D-1}x_{p}(m)\cdot e^{-j\frac{2\pi }{D}kp}=DFT\left[s_{p}(m)*h_{p}(m)\right]$$

In the above equation, DFT represents discrete Fourier transform, which can be replaced by fast Fourier transform (FFT) in hardware implementation to reduce hardware resource consumption and improve system operation speed. The improved structure is shown in the following Figure 3. This structure eliminates the multiplier factors used in traditional structures, resulting in improvements in the data path and resource utilization.

### 2.4. Improved Channel Decision Module

The proposed joint-decision process for eliminating channel ambiguity in the channelized receiver involves “Instantaneous Feature Extraction + Auto-Correlation Threshold Amplitude Detection + Phase Differential Instantaneous Frequency Measurement”. In the instantaneous feature extraction step, the coordinate rotation digital computer (CORDIC) algorithm is used to extract the instantaneous amplitude and phase information of the channelized output signal in vector mode. Traditional amplitude detection methods compare the signal amplitude with a fixed threshold value to determine the presence of signals in the channel [21]. In this study, the auto-correlation amplitude detection algorithm is used, which compares the signal amplitude information with dynamically changing threshold values that depend on the amplitudes of multiple points, resulting in more accurate detection results [22]. The phase differential instantaneous frequency measurement utilizes the extracted instantaneous phase information to measure the instantaneous frequency of the signal and, based on the frequency measurement result and the corresponding relationship with the channel center frequency, the correct channel output is determined, eliminating channel ambiguity. Figure 4 illustrates the joint-decision process, which ultimately yields accurate channel output results.

The CORDIC (coordinate rotation digital compute) algorithm calculates the amplitude and phase information of the output signal through iterative rotations [23]. The solving process is shown in Equation (8):(8)$$\begin{array}{l} \alpha_{k}(n)=\sqrt{I_{k}{}^{2}(n)+Q_{k}{}^{2}(n)}\\ \varphi_{k}(n)=\arctan\left[\frac{Q_{k}(n)}{I_{k}(n)}\right]\\ f_{k}(n)=\frac{f_{s}}{2\pi }\left[\varphi_{k}(n)-\varphi_{k}(n-1)\right] \end{array}$$

$I_{k}\left(n\right)$ and $Q_{k}\left(n\right)$ represent the real and imaginary parts of the output of the kth subchannel, and $\alpha_{k}\left(n\right)$, $\varphi_{k}\left(n\right)$, and $f_{k}\left(n\right)$ represent the instantaneous amplitude, phase, and frequency of the signal, respectively. The threshold value for adaptive detection is set based on the amplitude information of the output signal, and it is determined according to Equation (9):(9)$$V_{T}\left[n\right]=\beta \cdot \mu +\gamma =\beta \cdot \frac{1}{D}\cdot \sum_{k=1}^{D}A_{k}[n]+\gamma$$
where *β* represents the threshold coefficient, $A_{k}\left[n\right]$ represents the amplitude of the output signal from the kth subchannel, D represents the extraction factor under critical extraction state, $V_{T}\left[n\right]$ represents the adaptive detection threshold, and γ represents the noise floor introduced by the receiver. The amplitude value of the channelized output signal varies dynamically with the input signal, so the threshold values determined according to Equation (9) are also dynamic.

The instantaneous frequency of the output signal can be obtained using the phase difference instantaneous frequency measurement method, where the relationship between instantaneous frequency and instantaneous phase is given by Equation (10):(10)$$f_{k}(n)=\frac{\varphi_{k}(n)-\varphi_{k}(n-1)}{2\pi T}$$

T represents the sampling period of the signal. The frequency of the output signal can be obtained using the four-point phase difference averaging method to reduce measurement errors caused by noise, as the phase difference instantaneous frequency measurement method is sensitive to noise. In this design, the frequency of the output signal is given by the result obtained from the four-point phase difference averaging method.
(11)$$f_{k}(n)=\frac{\sum_{i=0}^{3}\Delta \varphi (n)}{8\pi T}$$

The entire decision-making process is as follows:

Set the detection threshold $V_{T}\left[n\right]$.Use the signal amplitude calculated from the CORDIC module as input. To reduce the impact of the signal-to-noise ratio, consider a signal to be present in the channel if the amplitude values $A_{k}\left[n\right]$ are greater than the detection threshold for five consecutive times. Then, proceed to the channel decision-making part.Finally, perform a phase-difference-based instantaneous frequency estimation for the channel. The frequency deviation value $\Delta f_{k}\left(n\right)$ represents the frequency deviation of the signal from the center of the channel. If $\left|\Delta f_{k}\left(n\right)\right|$ is less than half of the bandwidth, B, of the channel, it is considered that the signal is within the current channel, otherwise it is considered a false signal.

## 3. Simulation and Implementation of Improved Structure

### 3.1. Receiver System Simulation

Simulation Conditions: In MATLAB, the input signal is set as the sum of four signals from Table 1. The input signal-to-noise ratio (SNR) is set to 10 dB, with a total of 4096 sampling points. The number of digital channelized channels is 16. The prototype filter is set to have an order of 192 and a stopband attenuation of 60 dB. The frequency range for each sub-channel is shown in Table 2.

The channelized output results are shown in Figure 5. The simulation experiment tested the receiving capability of the channelized receiver for different input signals. The parameters of the four input signals were set in a controlled manner, selecting four typical frequencies to ensure that each signal belonged to a different channel. From Figure 5, it can be observed that the receiver model was able to correctly channelize and receive the superimposed signals from the four different input signals.

For the channels with detected signals, the extracted amplitude, phase, and frequency information are shown in Figure 6 and Figure 7 (taking channel 3 and channel 14 as examples). From left to right and top to bottom in Figure 6, we have the following results:

Amplitude extraction result for channel 3, Phase extraction result for channel 3, Amplitude extraction result for channel 14 and Phase extraction result for channel 14. From left to right in Figure 7, we have the following results: Frequency measurement result for channel 3 and Frequency measurement result for channel 14.

### 3.2. Hardware Implementation

The hardware implementation of the wideband digital channelized receiver primarily focuses on the large-scale and high-speed digital signal processing. The hardware platform is based on the Zynq UltraScale + RFSoC series chip, specifically the ZU27DR, which includes high-speed ADC and FPGA sections. The improved scheme of the entire receiver system is shown in Figure 8.

### 3.3. Data Extraction and Routing Module

In the HLS platform, when implementing the data extraction and routing module, input data can be stored in a multi-dimensional array based on the storage format of the data stream. During the transformation process, the input data needs to be rearranged by columns for parallel computation. Optimization directives can be added during the C Synthesis stage to reduce system latency [24]. Since the four loops in the data extraction and routing module have clear boundaries, and the input signal is divided into I and Q paths, there is no dependency between the real and imaginary part computations. Therefore, optimization can be achieved using the “Pipeline + Rewind” approach, as shown in Figure 9, to improve the parallel processing capability of the hardware.

### 3.4. Polyphase Filtering Module

In HLS, when designing the polyphase filtering module, the prototype filter coefficients need to be stored in an array. The coefficients are stored in an array of size (16, (filter order/16)). The implementation process of polyphase filtering is similar to convolution in the time domain, with the only difference being that FIR filtering requires zero padding at the end of the input sequence and uses a circular right shift to align the first item of the input sequence with the first item of the coefficient. Each right shift requires a multiplication operation, and the multiplication results are stored in registers. The final accumulated result is the output of the FIR filter. The polyphase filtering module consists of 16 parallel channels, and since the structure of each channel is the same except for the coefficients and input data, the implementation process of any sub-channel is shown in Figure 10.

### 3.5. The Fully Parallel FFT Module

Due to the high throughput requirements of the data output from the multi-phase filtering structure during FFT computation, a parallel FFT module that can perform parallel computation is needed [25]. However, the FFT IP core provided by Xilinx requires at least 16 clock cycles to complete a 16-point DFT computation [26], which does not meet the design requirements. In this paper, a parallel FFT structure is designed using a pipelined design approach, based on the radix-2 decimation in time (DIT) method, to compute the 16-point DFT. The core of the FFT algorithm is the butterfly operation, which divides the 16-point DFT into two 8-point DFTs based on the parity of the input sequence x(n), and further divides the 8-point DFTs into two 4-point DFTs, and so on, until the 16-point DFT is transformed into multiple 2-point DFT computations [27,28]. The butterfly operation flow of the 16-point parallel FFT module is shown in Figure 11.

### 3.6. Channel Decision Module

The CORDIC algorithm has three hardware implementation architectures: serial architecture, parallel architecture, and parallel pipeline architecture. These three architectures are based on the same basic structure of processing units, but operate at different shift amounts and storage angles, resulting in different implementation methods. In this paper, the parallel pipeline architecture is chosen for its design, which consumes significantly more hardware resources compared to the serial architecture.

By adding pipeline registers between each iteration, the processing speed of the system can be effectively improved, reducing the critical path length from N processing units in the parallel architecture to 1 processing unit in the pipeline architecture.

After obtaining the phase of the complex signal through the CORDIC module, the instantaneous frequency of the signal is calculated using the phase-difference-based frequency estimation algorithm, as shown in Figure 12.

## 4. Simulation and Analysis

The ChipScope, an integrated logic analyzer provided by Xilinx, was used to observe the signal, as described in [29]. In MATLAB, a sine wave signal with a carrier frequency of 935 MHz was generated and simulated. Since the input test signal is a complex signal, one path was selected for extraction and comparison verification in the real and imaginary parts. Here, the I path was selected for comparison verification. The first 10 columns of the extracted simulation results in MATLAB are shown in Figure 13.

The data shown in Figure 13 are obtained after quantization. The timing simulation results of the extraction path module are shown in Figure 14. After comparing with the simulation results in Matlab, it can be confirmed that the function of the module is correct.

The MATLAB simulation result and timing simulation result of the polyphase filter module are shown in Figure 15 and Figure 16, respectively. After comparison, it was found that the simulation results in HLS were correct, thus completing the consistency verification between the multi-phase filter module and the theoretical model.

The final channelized output result is shown in the Figure 17, where PFFT_OUT_3I is the real component of the third channel output, and PFFT_OUT_3Q is the imaginary component of the third channel output. It can be seen that there is an IQ component output in the third channel, proving that the calculation result of this module in FPGA is correct.

In addition, a chirp signal with a center frequency of 875 MHz and a bandwidth of 10 MHz was generated for the simulation of instantaneous feature extraction. The simulation results, depicted in Figure 18 and Figure 19, demonstrate the improved channel decision module’s ability to accurately extract amplitude, phase, and frequency information from the received signal. By comparing the extracted signal information with the sub-channel center frequencies defined in the receiver, false signals can be eliminated, resulting in a blind-zone-free reception in the receiver.

## 5. Conclusions

In this paper, an improved channel decision method has been proposed to address the issue of false signals in communication receivers based on the theory of blind-spot-free reception in a polyphase filtering structure. The method combines instantaneous feature extraction, the adaptive detection of signal amplitude, and phase difference frequency measurement algorithms to transform the process of eliminating false signals into extracting amplitude, phase, and frequency information from the output signal. This new approach improves the discrimination of useful signals and is more easily implemented on FPGA. Additionally, the hardware implementation of the receiver is optimized for speed and latency using HLS high-level synthesis technology, which significantly shortens the development cycle compared to traditional FPGA development processes. The simulation results confirm the practical value of this method for wideband digital receivers.

## Figures and Tables

**Figure 1 micromachines-14-01091-f001:**
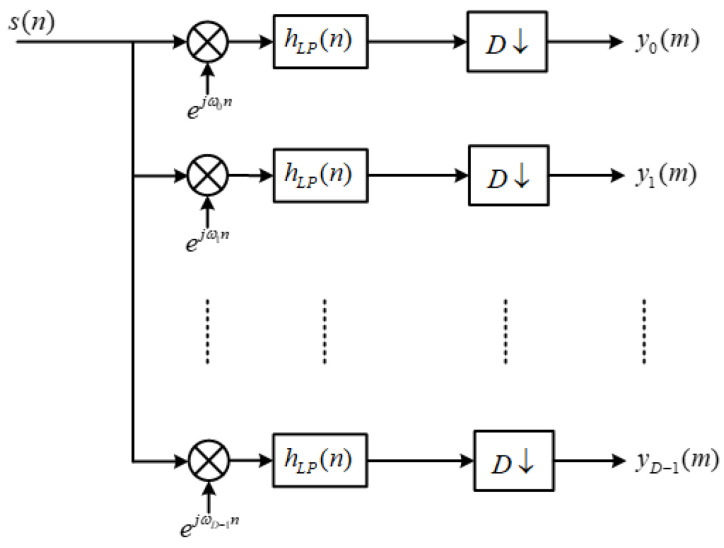
Original structure of digital channelized reception.

**Figure 2 micromachines-14-01091-f002:**
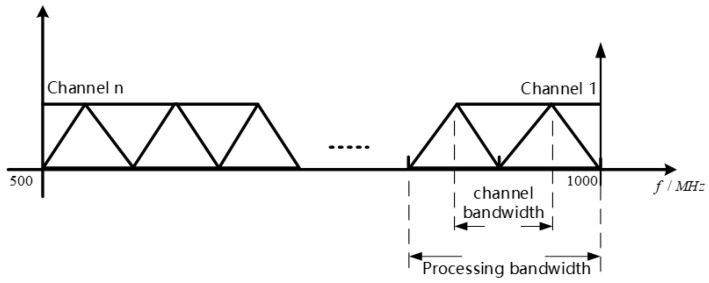
Adjacent channel with 50% overlapping uniform channel partitioning.

**Figure 3 micromachines-14-01091-f003:**
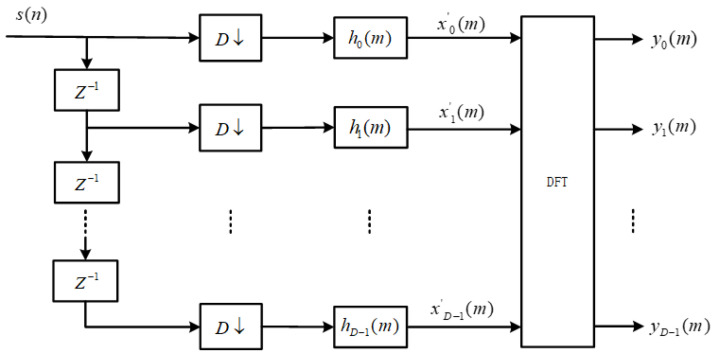
Efficient Digital Channelizer Receiver Model for Complex Signals.

**Figure 4 micromachines-14-01091-f004:**
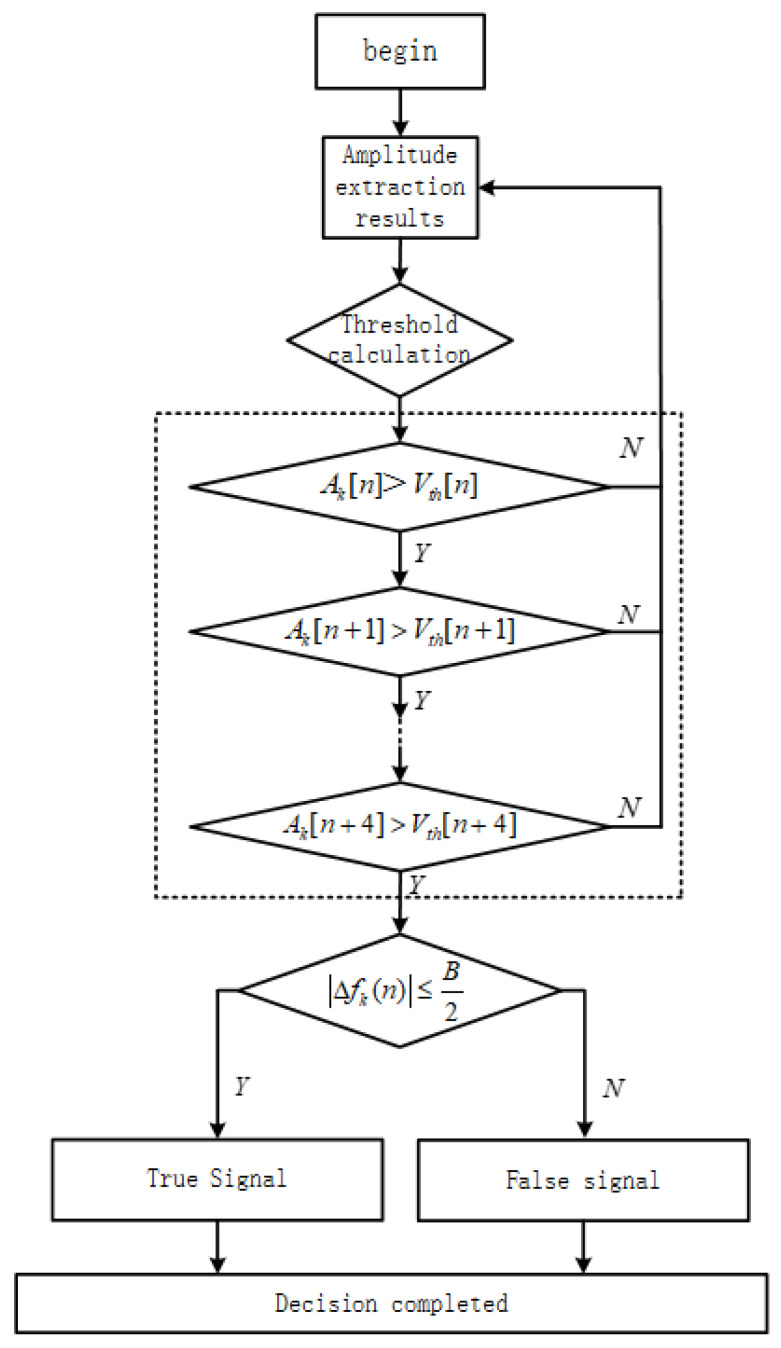
Joint-decision process.

**Figure 5 micromachines-14-01091-f005:**
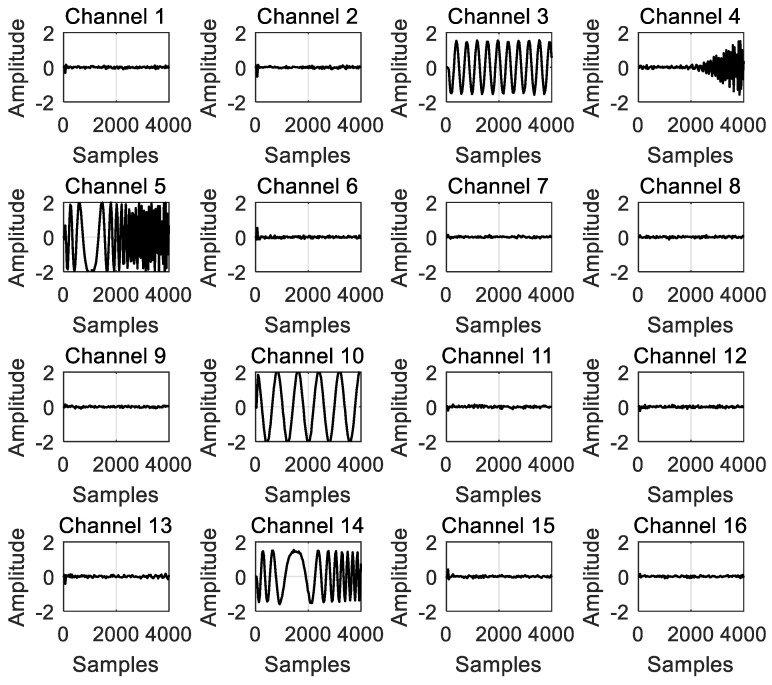
The channelized output results for 16 channels.

**Figure 6 micromachines-14-01091-f006:**
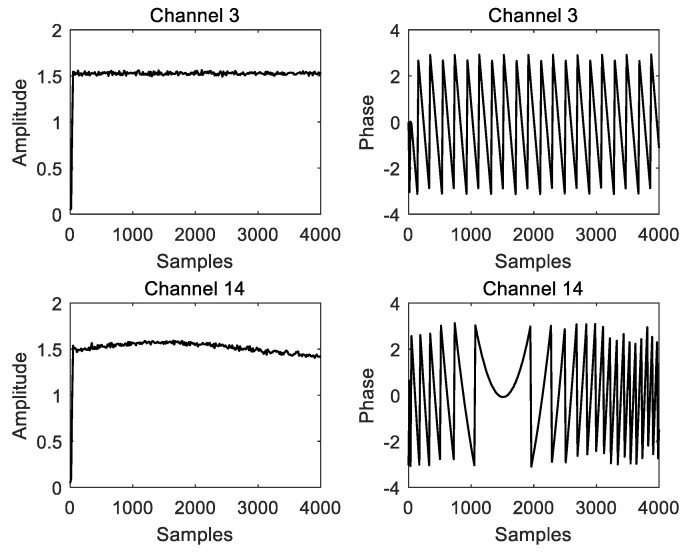
The amplitude and phase extraction results.

**Figure 7 micromachines-14-01091-f007:**
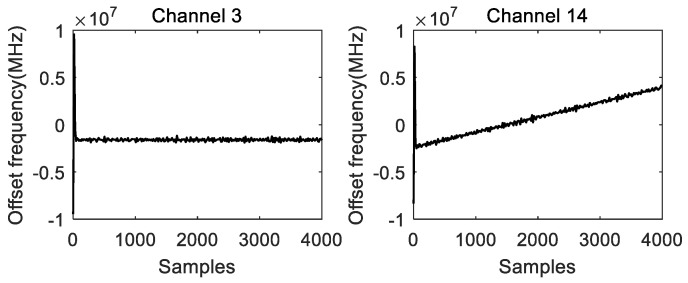
Instantaneous frequency measurement results.

**Figure 8 micromachines-14-01091-f008:**
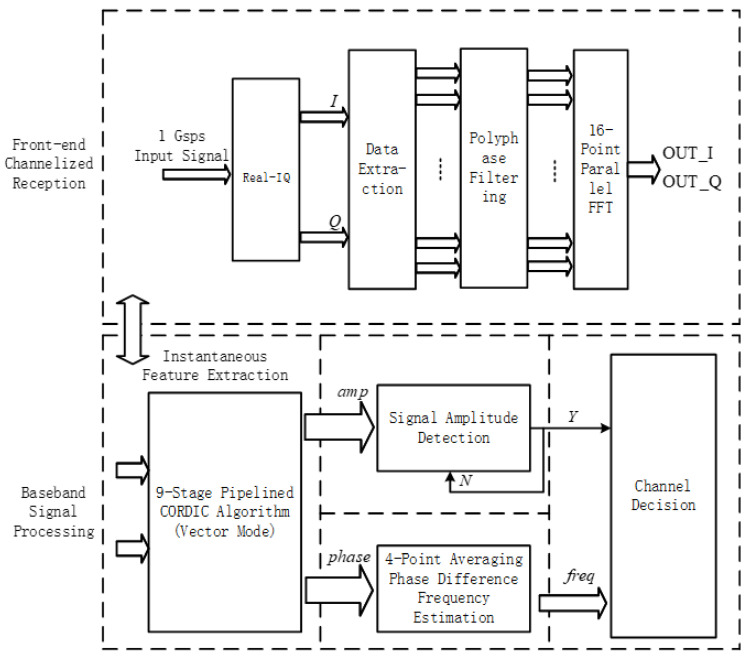
Improved Scheme for Wideband Digital Channelization Receiver.

**Figure 9 micromachines-14-01091-f009:**
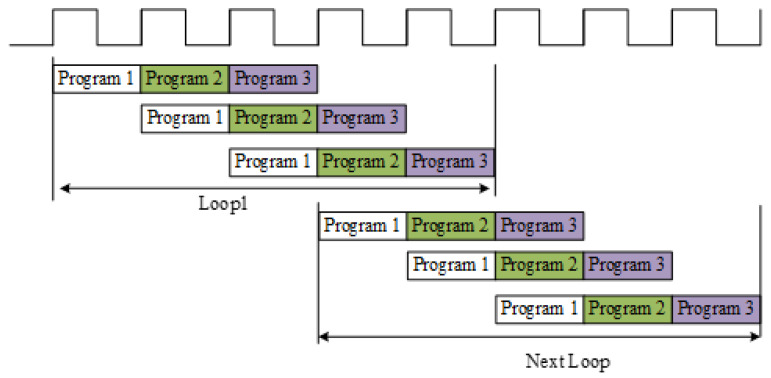
“Pipeline + Rewind” Optimization approaches.

**Figure 10 micromachines-14-01091-f010:**
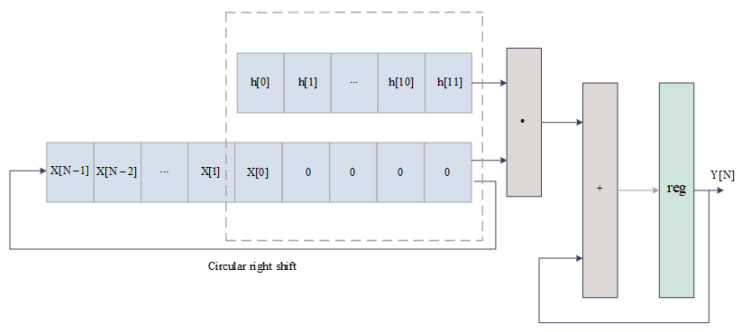
The implementation process of FIR filtering in any sub-channel.

**Figure 11 micromachines-14-01091-f011:**
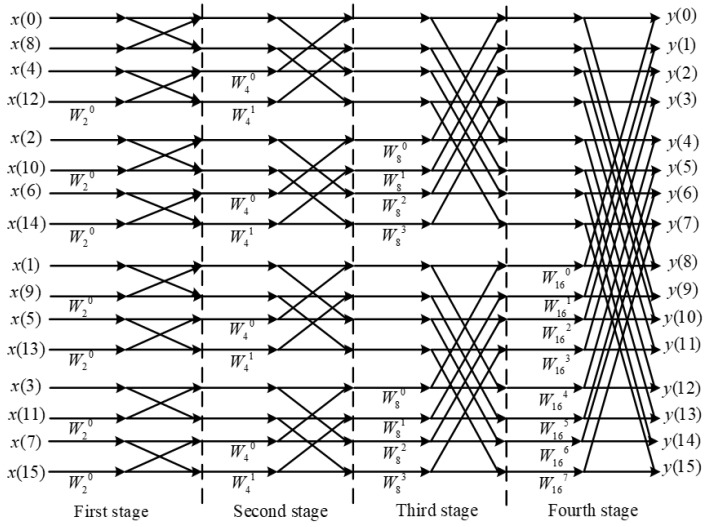
The 16-point FFT parallel processing flow.

**Figure 12 micromachines-14-01091-f012:**
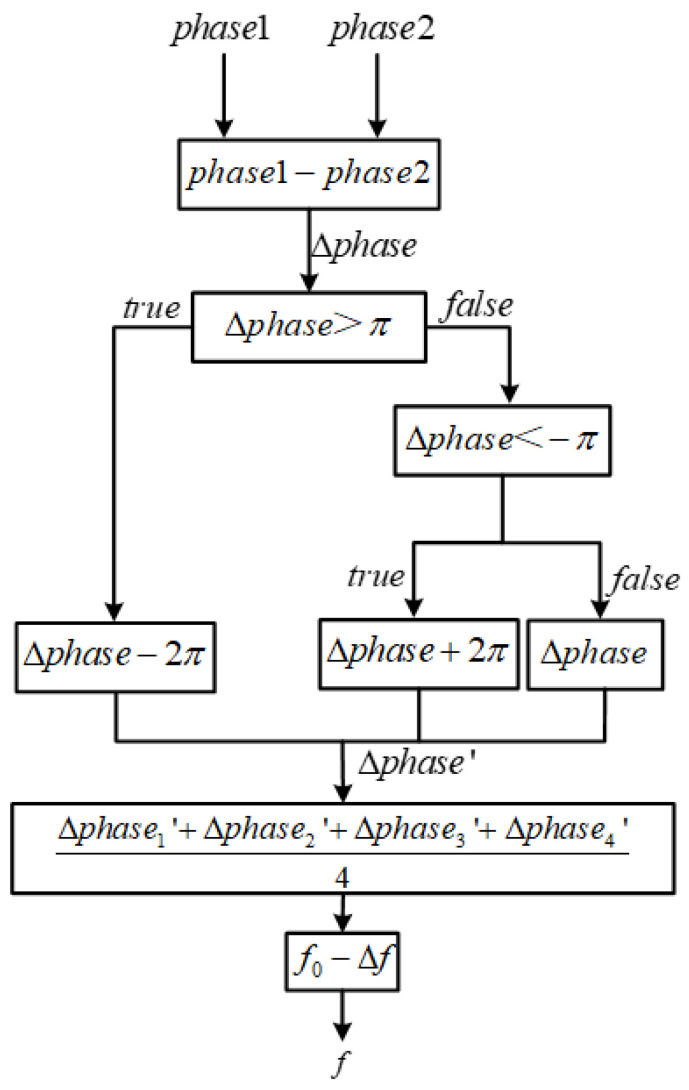
Instantaneous frequency measurement solution process.

**Figure 13 micromachines-14-01091-f013:**
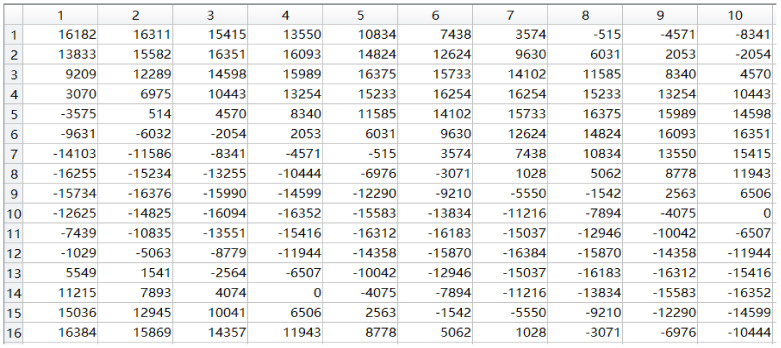
Simulation results of the extracted channel in MATLAB.

**Figure 14 micromachines-14-01091-f014:**
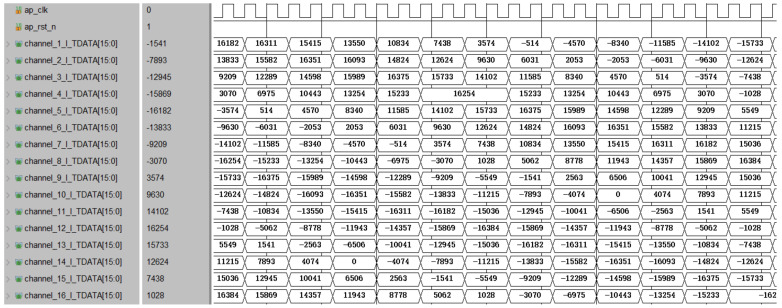
Time–domain simulation result of the extraction branch module.

**Figure 15 micromachines-14-01091-f015:**
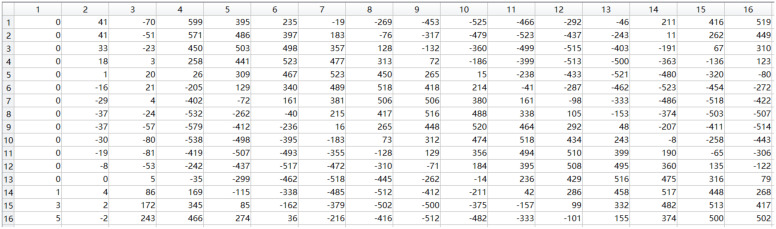
Polyphase filtering results in MATLAB.

**Figure 16 micromachines-14-01091-f016:**
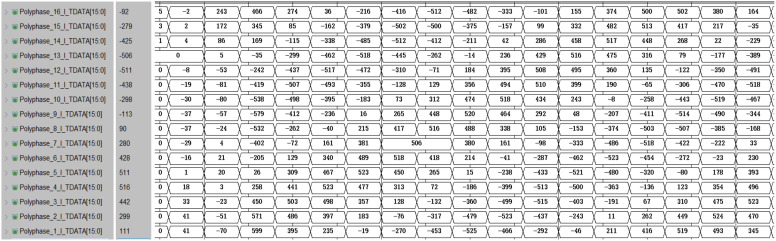
The timing simulation results of the polyphase filtering module.

**Figure 17 micromachines-14-01091-f017:**
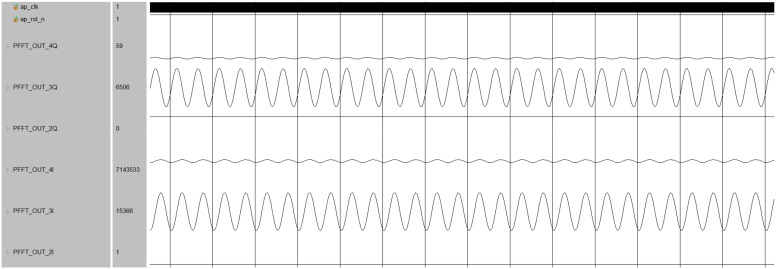
Time–domain simulation result of channelized output.

**Figure 18 micromachines-14-01091-f018:**
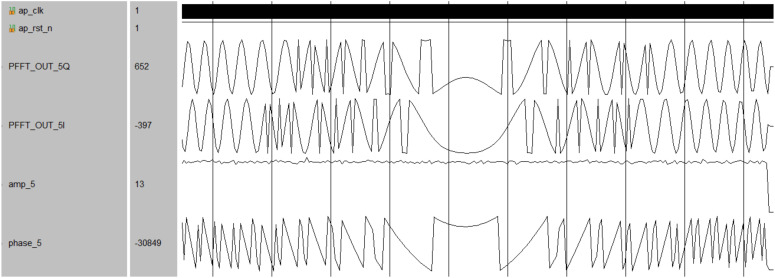
The results of amplitude and phase extraction.

**Figure 19 micromachines-14-01091-f019:**
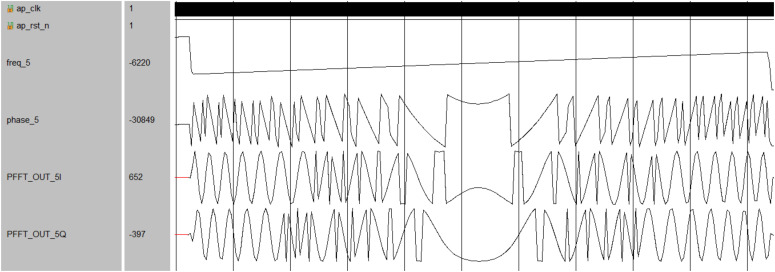
The results of instantaneous frequency extraction.

**Table 1 micromachines-14-01091-t001:** Information for setting up the overlapped input signals for simulation.

Input Signals	Frequency (MHz)	Amplitude (V)
Signal 1-sin	935	1.5
Signal 2-chirp	870–890	2
Signal 3-sin	720	2
Signal 4-chirp	590–600	1.5

**Table 2 micromachines-14-01091-t002:** Subchannel frequency range distribution table.

Channel Number	Channel Center Frequency (MHz)	Range of Subchannel Frequency (MHz)
1	1000	1000–984.375
2	968.75	984.375~953.125
3	937.5	953.125~921.875
4	906.25	921.875~890.625
5	875	890.625~859.375
6	843.75	859.375~828.125
7	812.5	828.125~796.875
8	781.25	796.875~765.625
9	750	765.625~734.375
10	718.75	734.375~703.125
11	687.5	703.125~671.875
12	656.25	671.875~640.625
13	625	640.625~609.375
14	593.75	609.375~578.125
15	562.5	578.125~546.875
16	531.25	546.875~515.625

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
