# Peer review of "An Improved Blind Zone Channelization Structure and Rapid Implementation Method"

_micromachines, 2023, doi:10.3390/mi14051091_

Round 1
Reviewer 1 Report
- The introduction section is too short. Please add some information about the background and related works and point out their drawbacks
- Please highlight the differences or improvement between other methods and yours
- The captions of Tables in this paper is missing.
- Page4, Line107: Please explain the CORDIC algorithm mentioned at the first time.
- Some references are considered to cite:
-
- Liu, X., Wang, Z. K., & Deng, Q. X. (2016). Design and FPGA implementation of a reconfigurable 1024-channel channelization architecture for SDR application. IEEE Transactions on Very Large Scale Integration (VLSI) Systems, 24(7), 2449-2461.
- IGARSS22-A Parallel Approach for Oil Palm Tree Detection on a SW26010 Many-Core Processor
- RS19-A real-time tree crown detection approach for large-scale remote sensing images on FPGAs
Author Response
No.2386190
12 May 2023
Title: An Improved Blind Zone Channelization Structure and Rapid Implementation Method
Dear editor/ reviewer,
We would like to thank you for your efforts in reviewing our manuscript titled " An Improved Blind Zone Channelization Structure and Rapid Implementation Method ", and providing many helpful comments and suggestions, which will all prove invaluable in the revision and improvement of our paper, as well as in guiding our research in the future.
We have studied your comments point by point, revised the manuscript accordingly. The amendments are highlighted in red in the revised manuscript.
We hope that the revised version of the manuscript is now acceptable for publication in Micromachines. If you have any queries, please do not hesitate to contact me.
Thank you again for your valuable comments and suggestions. I look forward to hearing from you soon in due course.
Yours sincerely,
Authors: Ziliang Jia *, Hongxia Liu *E-mails: 20111223127@stu.xidian.edu.cn, hxliu@mail.xidian.edu.cn
Response to reviewer:
We really appreciate you for your carefulness and conscientiousness. Your suggestions are really valuable and helpful for revising and improving our paper. There is a more detailed response letter with screenshots in the attached Word document. According to your suggestions, we have made the following revisions on this manuscript:
Comment 1:
The introduction section is too short. Please add some information about the background and related works and point out their drawbacks
Response 1:
Thank you for your insightful comment, Firstly, we expanded and supplemented the introduction section of the original manuscript, adding some introduction to the relevant background(Page 1, Lines 21-59), and providing information on previous works and their limitations (Page 2, Lines 60-68 and Page 3, Lines 83-87).
Comment 2:
Please highlight the differences or improvement between other methods and yours.
Response 2:
Thank you for your insightful comment, regarding the differences and improvements between the method proposed in this paper and other methods, they are supplemented in (Page 2, Lines 68-76) and (Page 3, Lines 87-96) in the article.
Comment 3:The captions of Tables in this paper is missing.
Response 3:
Thank you for the detailed review. I added a caption for Table 1 in (Page 8, Lines 227).
Comment 4:Page4, Line107: Please explain the CORDIC algorithm mentioned at the first time.
Response 4:
Thank you for the detailed review. I added a caption for Table 1 in (Page 6, Lines 168).
Comment 5:Some references are considered to cite:
1.Liu, X., Wang, Z. K., & Deng, Q. X. (2016). Design and FPGA implementation of a reconfigurable 1024-channel channelization architecture for SDR application. IEEE Transactions on Very Large Scale Integration (VLSI) Systems, 24(7), 2449-2461.
2.IGARSS22-A Parallel Approach for Oil Palm Tree Detection on a SW26010 Many-Core Processor
3.RS19-A real-time tree crown detection approach for large-scale remote sensing images on FPGAs
Response 5:
Thanks for your great suggestion on improving the accessibility of our manuscript.
Reference 1 is cited in (Page 12,Lines 295).
Reference 2 is cited in (Page 2,Lines 80).
Reference 3 is cited in (Page 3,Lines 82).
We would like to take this opportunity to thank you for all your time involved and this great opportunity for us to improve the manuscript. We hope you will find this revised version satisfactory.
Sincerely,
The Authors

Reviewer 2 Report
The paper presents an FPGA implementation for signal detection in a multichannel system. The proposal is interesting but some points must be addressed:
1) In section 3. The simulation signal requires a deeper explanation, including sampling frequency, and a time plot of the original signal. Why do figures only show 1000 points out of 4096?
2) Why in Figure 5 there are 5 channels with a signal while the original signal has four components?
3) I suggest including the frequency range for the plots in Figure 5.
4) A deeper discussion is needed in section 4, is there any way to compare Matlab simulation results with the hardware system? How do you quantitatively demonstrate the proper operation?
In addition, many typos must be corrected:
1) In line 99, figure3 should be Figure 3.
2) From lines 104 to 123, it uses a different font.
3) T, I, and Q are undefined.
4) Table 1 is misaligned.
5) There is a typo in 245 "ChirpScope" -> ChipScope.
6) Most references are outdated, only 4 out of 15 are under 5 years, and one of them is a reference manual from Xilinx that is not used in the work.
Author Response
No.2386190
12 May 2023
Title: An Improved Blind Zone Channelization Structure and Rapid Implementation Method
Dear editor/ reviewer,
We would like to thank you for your efforts in reviewing our manuscript titled " An Improved Blind Zone Channelization Structure and Rapid Implementation Method ", and providing many helpful comments and suggestions, which will all prove invaluable in the revision and improvement of our paper, as well as in guiding our research in the future.
We have studied your comments point by point, revised the manuscript accordingly. The amendments are highlighted in red in the revised manuscript.
We hope that the revised version of the manuscript is now acceptable for publication in Micromachines. If you have any queries, please do not hesitate to contact me.
Thank you again for your valuable comments and suggestions. I look forward to hearing from you soon in due course.
Yours sincerely,
Authors: Ziliang Jia *, Hongxia Liu *E-mails: 20111223127@stu.xidian.edu.cn, hxliu@mail.xidian.edu.cn
Response to reviewer:
We really appreciate you for your carefulness and conscientiousness. Your suggestions are really valuable and helpful for revising and improving our paper. There is a more detailed response letter with screenshots in the attached Word document.According to your suggestions, we have made the following revisions on this manuscript:
Comment 1:
In section 3. The simulation signal requires a deeper explanation, including sampling frequency, and a time plot of the original signal. Why do figures only show 1000 points out of 4096?
Response 1:
Thank you for your insightful comment, After inspection, it was found that the problem was caused by an erroneous assignment of the horizontal axis values during the graphing process. The horizontal axis value in the original image has been corrected to 4000. The reason for this is that in Matlab, when the horizontal axis is set to display up to 4096, the generated image will automatically be padded to 5000. Therefore, the last 96 points of data were discarded, which did not affect the results(Page 9, Lines 253-254,257-259).
Comment 2:
Why in Figure 5 there are 5 channels with a signal while the original signal has four components?
Response 2:
Thank you for the detailed review. This is because the signal in channel 4 is a false signal generated by the signal in channel 5. According to the relationship between the frequency distribution of the channels and the frequency of the signal, it can be seen that the 2-chirp signal will generate false signals in adjacent channels, which is the signal in channel 4.
Comment 3:I suggest including the frequency range for the plots in Figure 5.
Response 3:
Thank you for your useful suggestion. To make the frequency distribution of subchannels more intuitively displayed, it is listed in Table 2. This paper chooses 500-1000MHz for monitoring, and the subchannel bandwidth is calculated based on the receiver bandwidth and the number of subchannels.(Page 8,Lines 229-248).
Comment 4: A deeper discussion is needed in section 4, is there any way to compare Matlab simulation results with the hardware system? How do you quantitatively demonstrate the proper operation?
Response 4:
Thank you for the detailed review. I added time-domain simulation and Matlab result comparison for the data extraction branch and polyphase filter module in section 4. The data used in the simulation was quantized, and the comparison showed that the module simulation results were accurate. For the channelized output, the functionality can be directly judged by observing the waveform. According to the correspondence between the sub-channel frequency distribution and the input simulated signal frequency, it can be seen that the signal can be detected in channel 3, and there is no signal in other channels, indicating that the result is correct. Finally, I also conducted simulation on the instantaneous feature extraction function of the receiver system, and the result showed that the system could extract signal features completely and accurately, which was used to make the channel decision(Page 13,Lines 320-355).
Comment 5:In addition, many typos must be corrected:
Comment(1):In line 99, figure3 should be Figure 3.
Response(1):
Thanks for your great suggestion on improving the accessibility of our manuscript. I have corrected the error at this location.(Page 5,Lines 161)
Comment(2):From lines 104 to 123, it uses a different font.
Response(2):
Thanks for your great suggestion on improving the accessibility of our manuscript. I have corrected the error at this location.(Page 6,Lines 165-180)
Comment(3):T, I, and Q are undefined.
Response(3):
Thanks for your great suggestion on improving the accessibility of our manuscript. I have corrected the error at this location.(Page 7,Lines 186-190,Lines 202)
Comment(4):Table 1 is misaligned.
Response(4):
Thanks for your great suggestion on improving the accessibility of our manuscript. I have corrected the error at this location.(Page 8,Lines 227)
Comment(5):There is a typo in 245 "ChirpScope" -> ChipScope.
Response(5):
Thanks for your great suggestion on improving the accessibility of our manuscript. I have corrected the error at this location.(Page 13,Lines 320)
Comment(6):Most references are outdated, only 4 out of 15 are under 5 years, and one of them is a reference manual from Xilinx that is not used in the work.
Response(6):
Thanks for your great suggestion on improving the accessibility of our manuscript. I have updated and supplemented the original reference list, deleted the irrelevant references, and added a total of 29 references, including 16 references published within the last five years.(Page 13,Lines 320)
We would like to take this opportunity to thank you for all your time involved and this great opportunity for us to improve the manuscript. We hope you will find this revised version satisfactory.
Sincerely,
The Authors

Reviewer 3 Report
Review to the article No. 2386190
The proposed work analyzes the possibility of separating radio frequency signals and the technical possibility of separating "white" and "black" noise from each other in order to obtain a clean output signal. The article is quite relevant and can be published in the journal Micromachines. The authors propose a technical scheme for the implementation of such a possibility and on a specific example they explain the possible implementation of their idea.
I have only one comment that they need to address.
The list of references devotes too little space to sources in which the general principles of information transmission and anti-interference were previously outlined. The list presented in the article is rather scarce and requires replenishment.
After taking into account this remark the article can be published.
Author Response
No.2386190
12 May 2023
Title: An Improved Blind Zone Channelization Structure and Rapid Implementation Method
Dear editor/ reviewer,
We would like to thank you for your efforts in reviewing our manuscript titled " An Improved Blind Zone Channelization Structure and Rapid Implementation Method ", and providing many helpful comments and suggestions, which will all prove invaluable in the revision and improvement of our paper, as well as in guiding our research in the future.
We have studied your comments point by point, revised the manuscript accordingly. The amendments are highlighted in red in the revised manuscript.
We hope that the revised version of the manuscript is now acceptable for publication in Micromachines. If you have any queries, please do not hesitate to contact me.
Thank you again for your valuable comments and suggestions. I look forward to hearing from you soon in due course.
Yours sincerely,
Authors: Ziliang Jia *, Hongxia Liu *E-mails: 20111223127@stu.xidian.edu.cn, hxliu@mail.xidian.edu.cn
Response to reviewer:
We really appreciate you for your carefulness and conscientiousness. Your suggestions are really valuable and helpful for revising and improving our paper. There is a more detailed response letter with screenshots in the attached Word document. According to your suggestions, we have made the following revisions on this manuscript:
Comment 1:
The list of references devotes too little space to sources in which the general principles of information transmission and anti-interference were previously outlined. The list presented in the article is rather scarce and requires replenishment.
Response 1:
Thank you for your useful suggestion.
Thank you for your useful suggestion. I have updated and supplemented the references in this paper, increasing the total number of references to 29, including 16 references from the last five years.
We would like to take this opportunity to thank you for all your time involved and this great opportunity for us to improve the manuscript. We hope you will find this revised version satisfactory.
Sincerely,
The Authors

Round 2
Reviewer 1 Report
The authors have addressed all my issues.